# Type III TGF-β Receptor Down-Regulation Promoted Tumor Progression via Complement Component C5a Induction in Hepatocellular Carcinoma

**DOI:** 10.3390/cancers13071503

**Published:** 2021-03-25

**Authors:** Oscar Wai Ho Yeung, Xiang Qi, Li Pang, Hui Liu, Kevin Tak Pan Ng, Jiang Liu, Chung Mau Lo, Kwan Man

**Affiliations:** Department of Surgery, HKU-SZH & The University of Hong Kong, Pokfulam, Hong Kong SAR, China; why21@hku.hk (O.W.H.Y.); Qixiang515@connect.hku.hk (X.Q.); leepang@connect.hku.hk (L.P.); liuhui25@hku.hk (H.L.); ledodes@hku.hk (K.T.P.N.); u3003218@connect.hku.hk (J.L.); chungmlo@hku.hk (C.M.L.)

**Keywords:** clinical outcomes, tumor-suppressive protein, diagnosis prediction, therapeutic potential, complement component

## Abstract

**Simple Summary:**

The clinical implications of TGFβR3 downregulation are currently unknown in hepatocellular carcinoma (HCC). Clinically, we identified that HCC patients with low expression levels of tumoral TGFβR3 exhibited significantly late tumor stages and shortened survival outcomes. Moreover, HCC patients developed lower plasma levels of TGFβR3 (sTGFβR3) (8.9 ng/mL) compared to healthy individuals (15.9 ng/mL), which represented a potential diagnostic marker. Similar to tumoral TGFβR3, low levels of plasma sTGFβR3 are also associated with poor clinical outcomes in HCC. To determine its tumor-suppressing capacities, continuous injection of sTGFβR3 in an orthotopic liver tumor model was performed, resulting in 2-fold tumor volume reduction compared to control. Decreased expression of TGFβR3 induced the upregulation of tumoral complement component C5a in HCC, which was found to contribute to poor clinical outcomes and promote tumor progression via a novel function in activating the tumor-promoting macrophages.

**Abstract:**

Background and Aims—Transforming growth factor-beta (TGF-β) signaling orchestrates tumorigenesis and one of the family members, TGF-β receptor type III (TGFβR3), are distinctively under-expressed in numerous malignancies. Currently, the clinical impact of TGFβR3 down-regulation and the underlying mechanism remains unclear in hepatocellular carcinoma (HCC). Here, we aimed to identify the tumor-promoting roles of decreased TGFβR3 expression in HCC progression. Materials and Methods—For clinical analysis, plasma and liver specimens were collected from 100 HCC patients who underwent curative resection for the quantification of TGFβR3 by q-PCR and ELISA. To study the tumor-promoting mechanism of TGFβR3 downregulation, HCC mouse models and TGFβR3 knockout cell lines were applied. Results—Significant downregulation of TGFβR3 and its soluble form (sTGFβR3) were found in HCC tissues and plasma compared to healthy individuals (*p* < 0.01). Patients with <9.4 ng/mL sTGFβR3 exhibited advanced tumor stage, higher recurrence rate and shorter disease-free survival (*p* < 0.05). The tumor-suppressive function of sTGFβR3 was further revealed in an orthotopic mouse HCC model, resulting in 2-fold tumor volume reduction. In TGFβR3 knockout hepatocyte and HCC cells, increased complement component C5a was observed and strongly correlated with shorter survival and advanced tumor stage (*p* < 0.01). Interestingly, C5a activated the tumor-promoting Th-17 response in tumor associated macrophages. Conclusion—TGFβR3 suppressed tumor progression, and decreased expression resulted in poor prognosis in HCC patients through upregulation of tumor-promoting complement C5a.

## 1. Introduction

The multifunctional cytokine transforming growth factor-β (TGFβ) is a key regulator in multiple cellular processes including proliferation, differentiation, migration and immunological responses [1]. Unlike the other well-defined signaling pathways, whether TGF-β exhibits tumor-suppression or tumor-promoting functions remains controversial. Among the members of the super-family, type III receptor of TGF-β (TGFβR3) has shown a distinctive role in tumor biology. TGFβR3 is a co-receptor presenting TGF-β ligands to the TGFβR1 and ubiquitously expressed on nearly all cell types. It plays an essential role in mediating cell proliferation, apoptosis, differentiation, and migration in most human tissues [2]. Loss or reduced expression of TGFβR3 have been reported in many malignancies including breast, kidney, lung, ovaries, prostate and liver [3]. It has also been shown to be a key suppressor of tumor cell invasion, proliferation and angiogenesis in both in vitro and in vivo cancer models [3]. Apart from a transmembrane protein found in the cell membranes, the receptor can undergo ectodomain shedding from the cell surface to form soluble TGFβR3 (sTGFβR3) and be released to extracellular matrix and circulation. Studies have demonstrated its anti-tumor capacities in melanoma and breast cancer by sequestering TGF-β ligands for downstream pro-tumor signaling [4,5,6]. To date, little is known about the clinical implications and molecular mechanisms of TGFβR3 in hepatocellular carcinoma (HCC).

HCC has a very high metastatic and fatality rate (overall mortality to incidence ratio >90%), representing the second most common cause of death from cancer worldwide. Curative treatment options, including surgical and radiofrequency ablation, can be only applied to the patients with limited tumor burden [7]. Previously, we reported the clinical significances of alternative activated macrophages in promoting poor prognosis and tumor invasiveness in the disease [8]. In contrast, the recruitment and activating mechanisms of such an immune population by HCC remain elusive. TGFβR3 has been shown to possess the capacity to promote tumor-suppressing immunity [9]. Together with the observation that 66% patients showed decreased expression of the receptor, we investigated the clinical impact of TGFβR3 downregulation and its immuno-regulatory mechanism in HCC. In the present study, we revealed that the loss of TGFβR3 contributed to poor prognosis and promoted tumor progression via the upregulation of complement component C5a. 

## 2. Results

### 2.1. Aberrant Down-Regulation of TGFβR3 Expression in HCC Patients 

The clinical characteristics of the 100 patients who underwent curative resection were described in Appendix A. To determine the protein and gene expression level of tumoral TGFβR3, immunohistochemistry, Western blotting and quantitative-PCR were applied. Downregulation of TGFβR3 in tumor was consistently observed in both immuno-staining and blotting studies (Figure 1A–C). Further transcript analysis revealed a significant 1.43- and 0.89-fold decrease in TGFβR3 in HCC tumor compared to adjacent non-tumoral and normal tissues, respectively (Figure 1D). Such downregulation was observed in 66% of the studied HCC population, as well as in seven HCC-patient-derived cell-lines (Figure 1E). All the data collectively illustrated the significant reduction in TGFβR3 at both transcript and protein levels in HCC patients. Such down-regulation was also validated in publicly available datasets from Oncomine (www.oncomine.org) and TCGA/GTEx by using GEPIA (http://gepia.cancer-pku.cn/) (Appendix A).

### 2.2. Down-Regulation of TGFβR3 Correlated with Poor Prognosis in HCC Patients

When analyzed by the Kaplan–Meier method with log-rank statistics, low levels of tumoral TGFβR3 transcript were found to be associated with poor overall survival (*p* = 0.017) and disease-free survival (*p* = 0.047) (Figure 1F,G). Similar clinical traits of TGFβR3 were also confirmed in the publicly available datasets from GEPIA (Appendix A). Apart from survival outcomes, the correlation between clinicopathological characteristics and TGFβR3 expression was examined. As summarized (Table 1), high levels of alpha-fetoprotein (AFP) (>20ng/mL)(*p* = 0.014), and advanced tumor stage in all grading systems, including UICC (*p* = 0.01), Edmonson (*p* = 0.003) and AJCC (*p* = 0.048), were found to be associated with low expression of TGFβR3. These findings highlighted the clinical significance of TGFβR3 in prognosis as well as tumor progression in HCC patients. 

### 2.3. Soluble TGFβR3 (sTGFβR3) Exhibited Diagnostic and Prognostic Potentials in HCC

As mentioned, TGFβR3 undergoes ectodomain shedding released from tissue to extracellular matrix and circulation as soluble TGFβR3 (sTGFβR3). Compared to healthy individuals (15.4 ng/mL), a significant reduction in plasma sTGFβR3 was observed in 72% of HCC patients (8.9 ng/mL) (*p* < 0.01) (Figure 2A). Receiver-operating characteristics (ROC) curve analysis revealed that sTGFβR3 served as a biomarker for differentiating patients with HCC from healthy patients with an AUC of 0.838 (95% CI, 0.78 to 0.90) (*p* < 0.001) (Figure 2B). At the cut-off value of 9.4 ng/mL sTGFβR3 in plasma, the sensitivity was 82.7% and the specificity was 77.4%, respectively. With the acquired threshold value, Kaplan–Meier analysis revealed that patients with less than 9.4 ng/mL significantly developed poorer overall and disease-free survival compared to the ≥9.4 ng/mL group (Figure 2C,D, respectively) (*p* < 0.05). In terms of clinicopathological characteristics, a continuous decrease in plasma sTGFβR3 was observed in patients with advanced tumor stages (Stage I and II: 14.99 ng/mL ±3.62; Stage III: 7.67 ng/mL ± 0.71; Stage IV: 5.64 ng/mL ± 0.42) (Figure 2E). Low levels of plasma sTGFβR3 were also associated with high levels of bilirubin (>20μmol/l) (*p* < 0.01), large tumor size (>5cm) (*p* = 0.012) and advanced tumor stages UICC (*p* < 0.01) and AJCC (*p* = 0.017) (Table 2). Based on the finding that plasma sTGFβR3 exhibited similar clinical associations with tumoral TGFβR3, we further confirmed their strong correlation in HCC patients (R^2^ = 0.112, *p* < 0.01) (Figure 2F). 

### 2.4. TGFβR3 Treatment Suppressed HCC Tumor Growth In Vivo

With the evidence of advanced tumor stage patients having poorly expressed TGFβR3, its tumor-suppressive function was further studied in a nude mouse orthotopic liver cancer model. Weekly intraperitoneal injection of recombinant sTGFβR3 (25 μg per mouse) significantly reduced tumor density by bioluminescence, with a 2-fold decrease compared to the untreated group (276.9U ± 40.65 vs. 138.1U ± 29.1, *p* = 0.017) (Figure 3A-i). Consistently, the HCC tumor volume measured after scarification was also found to be 1.6-fold lower in the sTGFβR3 treatment group (0.96 cm^3^ ± 0.14 vs. 1.53 cm^3^ ± 0.2) (*p* = 0.037) (Figure 3A-ii). Further TUNEL assay demonstrated the increase in apoptotic tumor cells in the sTGFβR3 treatment group (Figure 3A-iii).

Apart from sTGFβR3, the tumor-suppressive role of TGFβR3 was examined in a subcutaneous tumor model in nude mice, induced by the HCC cell-line MHCC97L, which was over-expressed with TGFβR3 (MHCC97L-TGFβR3). Including MHCC97L-NTC as a negative control, both the non-transfected and transfected cell-lines were injected into the left and right flank of each mouse, respectively (Figure 3B-i). A significant decrease in tumor size in the MHCC97L+TGFβR3 (0.35 cm^3^ ± 0.07) group compared to MHCC97L-NTC (0.74 cm^3^ ± 0.11) was observed (*p* = 0.045) (Figure 3B-ii). 

### 2.5. Loss of TGFβR3 Induced the up-Regulation of C5a which Associated with Poor Prognosis in HCC

To simulate its loss during HCC progression, TGFβR3 was knocked out by transfection of a Crispr/Cas9 KO plasmid in two hepatic non-HCC cell lines MIHA and LO2. Through analysis by ELISA and molecular array study, we discovered a significant increase in complement component C5a secretion in both MIHA-TGFβR3 KO (0.833 ng/mL ± 0.083) and LO2-TGFβR3 KO (0.66 ng/mL ± 0.037) supernatants compared to vector control, parental MIHA-scramble (0.348 ng/mL ± 0.061) and LO2-scramble cells (0.23 ng/mL ± 0.03) (*p* < 0.01) (Figure 4A). Furthermore, high secretory levels of C5a in all HCC were identified (Figure 4B) with minimal co-expression with TGFβR3 (Appendix A) in contrast to MIHA and LO2. Interestingly, we detected human C5a in both tumoral tissue and plasma in an orthotopic nude mice model with MHCC97L-induced HCC tumor at week 4 (Appendix A). These demonstrated the capacity of continuous secretion of C5a by implanted low-TGFβR3-expression MHCC97L cells in vivo. Clinically, C5a-expressing HCC cells were also identified to be present in tumoral tissue (Appendix A). Quantitative analysis revealed the significant increase in the complement protein in patients’ tumoral tissue (Figure 4C-i) and plasma (Figure 4C-ii). Importantly, strong reverse relationships between C5a and TGFβR3 in both human liver tumor tissue (*p* = 0.0259) (Figure 4D-i) and plasma (*p* = 0.011) (Figure 4D-ii) were validated. Consistent with the clinical phenotypes of TGFβR3 downregulation, increased levels of plasma C5a were first detected in patients with advanced (stage III, 13.4 ng/mL ± 1.86 and IV, 13.8 ng/mL ± 3.003) compared to early (stage I/II, 7.428 ng/mL ± 1.075) tumor stage (*p* = 0.046) (Figure 4E). High levels of both tissue and plasma C5a were also observed to be strongly correlated with large tumor size (*p* < 0.01; *p* < 0.0198) (Figure 4F-i,ii). Patients with high levels of plasma C5a (>12 ng/mL) developed significantly shorter disease-free survival compared to the low-expression group (Log rank: 5.798, *p* = 0.016) (Figure 4F-iii). 

### 2.6. Complement C5a Activated the Th-17 Responses in Tumor Promoting Macrophages 

In myeloid cells, macrophage is a major target of C5a, and as we had previously reported its critical roles in HCC, the function of the complement protein in the immune subset was further investigated. First, a positive correlation between plasma C5a and alternatively activated tumor (M2) macrophage marker (scavenger receptor) was identified in HCC patients (R^2^ = 0.05, *p* = 0.030) (Figure 5A). When analyzed by human PCR array, incubation of 1μg/mL recombinant C5a significantly induced the Th-17-related cytokines (IL-17, IL-21 and IL-22) and IL-17 regulatory genes (CXCL-1, CXCL-2, CSF-2) in M2 macrophages, but did not classically activate the M1 subtype (Figure 5B)(Appendix A). Significantly increased levels of C5a receptor were also detected in both activated macrophage populations in response to the complementary component (Appendix A). Importantly, incubating M2 macrophages with HCC-cell-cultured medium (MHCC97L, MHCC97H, Hep3B, PLC and Huh7) containing high levels of C5a significantly elevated the expression level of IL-17 (Figure 5C). Furthermore, patients with high levels of plasma C5a exhibited up-regulation of IL-17-secreting macrophages in HCC tumor in both immunostaining (Figure 5C) and flow cytometry analysis (Figure 5D).

## 3. Discussion

Despite the predominant downregulation of TGFβR3 shown in both public databases and other studies [9], the clinical implication in HCC is unknown to date. For the first time, we revealed that decreased levels of tumoral and plasma TGFβR3 are strongly associated with advanced tumor stage and tumor size, and, more importantly, poor clinical outcome, including shortened overall and disease-free survival. Apart from its prognostic associations, high differential levels of soluble TGFβR3 in plasma between HCC patients and healthy individuals indicated the diagnostic potential, as validated by ROC curve analysis with the highly specific cut-off value of 9.4 ng/mL. All the evidence collectively highlighted the clinical significance of TGFβR3 downregulation in HCC patients. 

Based on the clinical evidence, the tumor-suppressive function of TGFβR3 in HCC was speculated and revealed in two tumor models. First, the continuous administration of recombinant sTGFβR3 significantly decreased HCC tumor size by 2-fold compared to controls in the orthotopic nude mice liver tumor model, with an increased level of tumor cell apoptosis. Similar antitumor activity of recombinant sTGFβR3 in tumor models was previously reported in breast and prostate cancer [4,6,10,11], indicating its effectiveness against different malignancies but not in HCC. Apart from sTGFβR3, restoring TGFβR3 in lowly expressed HCC cells also significantly reduced tumor growth by 2.1-fold in the subcutaneous tumor model. Findings from both models collectively illustrated the direct tumor-suppressive functions of TGFβR3 in HCC. Several studies indicate that TGFβR3 suppresses tumor development via negatively mediating TGF-β signaling [12,13,14]. In the present study, we identified the significant up-regulation of phospho-SMAD2/3 in tumoral tissue with minimal expression levels of TGFβR3 (Appendix A). 

The mechanisms of TGFβR3 dysregulation remain unknown and it is tempting to correlate them with tumor hypoxia based on the close relationship between hypoxia-inducing factor 1 alpha (HIF-1α) and the TGF-β signaling pathway. Nevertheless, emerging evidence has shown that loss of TGFβR3 is related to paracrine or cell autonomous signaling, resulting in the alteration in the tumor immune environment, including dendritic cells [6]. Apart from cellular mechanism, recent studies reported that the disruption of TGFβR3 induced the dysregulation of the complement components, including C4a and the complement factor D in breast and prostate cancer [5,15]. Since the liver possesses many unique immunological properties, as the residence of many immunological cells and the synthesis site of numerous innate proteins, including complement components, we speculated an induced tumor-promoting immunological mechanism following the dysregulation of TGFβR3 in HCC. More importantly, we previously reported that alternatively activated (M2) macrophages represented the key immune population contributing to HCC progression [8]. Hence, we focused on studying the secretory profiles of TGFβR3-down-regulated cells to identify potential immune-regulatory mechanisms associated with macrophages in HCC. By silencing TGFβR3 in normal hepatocytes, we observed elevated secretion of one particular complement component, C5a, but not C3a and C4a. Further studies confirmed that HCC cell-lines with low TGFβR3 expression also displayed high secretory level of C5a. 

Apart from being the central chemo-attractant provoking an innate immune response, emerging evidence has also suggested the novel roles of C5a in shaping the tumor immune microenvironment [16,17]. Most tumors are rich in complement proteins, particularly C3a and C5a, produced directly by cancer cells, which have a variety of tumor-promoting mechanisms without activating the complement cascade [18,19,20,21,22]. Clinically, the concentration of both plasma and tumoral C5a determine the disease progression in malignancies, including the lung and ovaries [18,23,24]. Noteworthy hepatocytes are responsible for biosynthesizing the majority of complement proteins. Despite the close relationship between hepatocyte and C5a, its clinical implications and underlying mechanisms in HCC are currently unclear. For the first time, we showed that HCC cells upregulated the secretion of C5a, which is particularly induced by the downregulation of TGFβR3. High levels of the complement protein were associated with late tumor stage, increased tumor size and poor disease-free survival in HCC patients, consistent with the clinical associations of decreased TGFβR3 expression. On the other hand, both the tumoral and circulatory levels of C5a were found to be highly dependent on both tumoral and soluble TGFβR3 in reverse manners. The clinical and in vitro evidence collectively suggested that dysregulation of TGFβR3 in hepatocyte and HCC cells activated the secretion of pro-tumoral C5a in HCC. 

C5a possess many immuno-modulatory functions, particularly in the innate immunity [17,19]. Direct cytokine and chemokine production in the immune cells are also found to be regulated by the complement protein [25]. Here, we discovered the novel role of C5a in enhancing the tumor-promoting phenotypes on the alternatively activated (M2) macrophage. The treatment of either HCC cells supernatants with rich-in-C5a or recombinant C5a to M2 macrophages significantly induced its expression of C5aR and, surprisingly, IL-17 regulatory genes (CXCL-1, CXCL-2, MCSF, IL-17F) and Th-17 secretory cytokines (IL-17F, IL-21 and IL-22). In contrast, a minimal effect of C5a in another subtype, classically activated (M1) macrophage, was observed. Numerous studies suggested the clinical significance of Th-17 responses in contributing to poor survival outcome in HCC patients by the pro-tumor functions of IL-17, IL-21 and IL-22, including tumor proliferation and angiogenesis [26,27,28,29,30]. Importantly, the capability of HCC-stroma-associated macrophages in inducing Th-17 T cell response was shown [31]. The findings in the present study collectively indicated a novel mechanism of C5a in activating Th-17 tumor-promoting phenotypes in M2 macrophages. 

In conclusion, we first reported that both the downregulation of TGFβR3 and increased C5a are associated with poor clinical outcomes in HCC. Plasma sTGFβR3 served as a potential diagnostic biomarker for identifying patients with advanced tumor stages. A novel pro-tumoral mechanism of TGFβR3 downregulation via C5a-activated tumor-promoting macrophages was revealed. Further applications of sTGFβR3 and the C5a inhibitor may represent a new approach to treating HCC patients.

## 4. Materials and Methods 

### 4.1. Patient Samples

Liver tumor tissues and blood samples were randomly collected from 100 patients (aged 3–83 years, 77% male) who underwent curative surgery for HCC in Queen Mary Hospital from 2004 to 2008. Normal liver tissues were from a healthy living donor (*n* = 100). Ethical approval (UW11–100 (HKU/HA HKW IRB)) was obtained from the University of Hong Kong and Hospital Authority-Hong Kong Western Cluster and consent was signed by the studied patients.

### 4.2. Orthotopic Nude Mice Liver Tumor Model with sTGFβR3 Treatment

Male athymic nude mice (BALB/c nu/nu, 4–6 weeks old) (total number = 20) were used and all the studies were conducted according to the Animals (Control of Experiments) Ordinance (Hong Kong) and the Institute’s guidance on animal experimentation. All mice were housed in a pathogen-free animal facility at 22 ± 2 °C under controlled 12-h light/dark cycles. Mice were given regular chow (5053-PicoLab^®^ Rodent Diet 20, Lab Diet, MO, USA) and had access to autoclaved water. Surgical procedures were as described previously [32]. Briefly, 3 × 10^5^ MHCC97L suspended in 0.2 mL DMEM were injected subcutaneously into the flanks of mice. After 4 weeks, the subcutaneous tumors were resected and diced into 1 mm^3^ cubes, which were then implanted in the left lobes of the livers of another group of nude mice. Simultaneously, 2 mg/kg of recombinant sTGFβR3 were injected peritoneally in the treatment group weekly for four weeks. For the number of mice applied for the experiment, seven were used for both the treatment groups and as a negative control. Mice injected with PBS served as a negative control. Tumor size and metastasis of MHCC97L xenograft were monitored weekly by Xenogen IVIS^®^ (Xenogen IVIS^®^ 100, Caliper Life Sciences, Hopkinton, MA, USA). All mice were sacrificed at week 5 and the size of liver tumor was measured.

### 4.3. RT2 Profiler PCR Array

Total RNA was extracted from C5a treated (1 μg/mL, 12 h) and untreated M1 and M2 macrophages, using the TRIzol reagent (Invitrogen) and purified with the RNAeasy MinElute Cleanup Kit (Qiagen) according to standard protocols and converted to first strand cDNA using RT2 First Strand Kit (Qiagen). Gene expression of 84 cytokines and chemokines-related genes was analyzed using the Human Cytokines and Chemokines RT2 Profiler™ PCR Array (PAHS-150Z, Qiagen) according to the manufacturer’s protocol.

### 4.4. In Vitro Over-Expression and Knockout of TGFβR3 

TGFβR3-deficient non-HCC hepatocytes LO2 and MIHA were established using a CRISPR/Cas9 system with the plasmid TGFβR3 CRISPR/Cas9 KO Plasmid (SC-401316) purchased from Santa Cruz Biotech. Control CRISPR/Cas9 Plasmid (SC-418922) was applied as a negative control. All the cells were transfected using Lipofectamine 2000 (Life Technologies) for 48 h with 3 μg TGFβR3 and control plasmid. The transfection efficiency was determined by fluorescent microscopy and cells were sorted by FACS analysis. On the other hand, to over-express TGFβR3 in HCC MHCC97L cell line, a TGFβR3 human clone, the pCMV6-AC-GFP vector, was purchased from Origene Technologies. 

### 4.5. In Vivo Study of Tumorigenicity of TGFβR3 Over-Expressed Cells

Male athymic nude mice (BALB/c nu/nu, 4–6 weeks old) were used. For the xenograft tumor growth assay, control cells (MHCC97L-NTC) were injected subcutaneously into the left dorsal flank of mice, and TGFβR3-expressing cells (MHCC97L-TGFβR3) were injected into the right dorsal flank of the same animal. Tumor formation in nude mice was monitored over a 4-week period, and the tumor volume was measured weekly and calculated as 0.5 × L × W2. The mice were euthanized on the fifth week, and the tumors were excised and embedded in paraffin. Sections (5 μm) of tumors were stained with H&E to visualize the tumor structure.

### 4.6. Quantitative Real Time RT-PCR 

Total RNA was extracted from cell lines and frozen tumor specimens using Trizol Reagent (Invitrogen). Total RNA was reverse-transcribed with High-Capacity cDNA Reverse Transcription Kit (Applied Biosystems). Messenger RNA expression levels were determined by real-time PCR using Fast Start SYBR Green Master (Roche Diagnostic) with an ABI Prism 770 sequence detection system. Primers used for the amplification of human genes were as follows.

Transforming growth factor beta type III receptor (TGFβR3) forward: 5′–ACC GTG ATG GGC ATT GCG TTT GCA–3′, reverse: 5′–GTG CTC TGC GTG CTG CCG ATG CTG T–3’. C5aR forward: 5′–GAG CCC AGG AGA CCA GAA CAT G–3′, reverse: 5′–TAC ATG TTG AGC AGG ATG AGG GA–3′. Class A macrophage scavenger receptor (SA) forward: 5′–GCA GGG CCC TCT TAA GAT CA –3′, reverse: 5′–AAC ACG GGA ACC AAA GTC AT–3′. Interleukin 17 (IL-17) forward: 5′–TCC CAC GAA ATC CAG GAT GC–3′, reverse: 5′–GGA TGT TCA GGT TGA CCA TCA C–3′. 18s forward: 5′–CTC TTA GCT GAG TGT CCC GC–3′, reverse: 5′–CTG ATC GTC TTC GAA CCT CC–3′. For clinical tissue samples, the relative fold change of the target gene expressed in intratumor and peritumor tissue was determined by the following equation: 2–ΔΔCt (ΔCt = ΔCttarget—ΔCt18s; ΔΔCt = ΔCt—ΔCtnormal) which was normalized to the average fold change in the normal liver tissues defined as 1.0. TGFβR3 expression level in each case was classified as either a high or low group determined by the ROC curve. For in vitro study, all experiments were performed in triplicate and repeated three times. 

### 4.7. Immunostaining and Flow Cytometry

Immunohistochemistry and flow cytometry were applied to examine the level of different proteins (TGFβR3, CD163, C5a, IL-17 and glypican) in clinical specimens. For immuostaining, paraffin-embedded liver and liver tumor were sectioned to 4 μm thickness. Endogenous peroxidase blockage and visualization were achieved with Dako Envision+ System (Dako). Following heat-induced epitope retrieval according to the manufacturer’s protocol, incubation with antibodies against human TGFβR3 (1:100, Abcam), CD163 (1:100, Leica Novocastra), C5a (1:200, Abcam), IL-17 (1:100, R&D Systems) and glypican (1:100, Abcam) were performed. All immunohistochemistry and immunofluorescence images were taken by the Niken ELIPSE E600 and Invitrogen EVOS FL Color Imaging System, respectively. All flow cytometry analyses were performed by CytoFlex S Beckman Coulter. 

### 4.8. ELISA

The level of soluble TGFβR3 (Sigma) and complement component C5a (BD Biosciences) were quantified in patients’ plasma and tumoral tissue. All procedures were performed according to the manufacturer’s instructions.

### 4.9. Cell Culture and Stimulation 

Authentication for the MHCC97L, MIHA and LO2 cells used in the present study was performed using Powerplex 16 HS kit (Promega) after PCR amplification and capillary electrophoresis. The human acute monocytic leukemia cell line THP-1 was purchased from ATCC and maintained according to ATCC guidelines. Other hepatic and HCC cell-lines were purchased or obtained as described in the Appendix A. The protocols of M1 and M2 macrophage polarization from THP-1 were adopted from Tjiu et al. [33]. Briefly, to induce M1-polarized phenotype, 25 ng/mL interferon gamma (IFN-γ; Invitrogen) and 150 ng/mL lipopolysaccharide (LPS; Sigma) were added to 1 × 10^μ^ THP-1. To induce M2-polarized phenotype, 20 ng/mL of recombinant interleukin 4 (IL-4; Invitrogen) and recombinant interleukin 13 (IL-13; R&D Systems) were used. All stimulation lasted 24 h at 37 °C, and culture was washed thoroughly with PBS three times prior to further study.

For luciferase-labeling, MHCC97L were transfected with luciferase gene in pGL3 vector (Promega), and positive clones were selected according to luciferase activity in Xenogen In Vivo Imaging System 100 (Xenogen IVIS^®^ 100, Xenogen Corporation).

### 4.10. Statistical Methods

Comparisons and correlations of quantitative data between the two groups were analyzed by unpaired Student’s *t*-test and chi-square test, respectively. Categorical data were analyzed by Fisher’s exact test. The Cox proportional hazards model was applied to determine the independent factors of survival, based on the variables selected on univariate analysis. The log-rank test for comparison of survival in Kaplan–Meier survival plot was used for analysis. A *p* < 0.05 was considered statistically significant. All analyses were performed with Graphpad Prism 5.0 and SPSS18.0.

## 5. Conclusions

In conclusion, we first reported that both the downregulation of TGFβR3 and increased C5a were associated with poor clinical outcome in HCC. Plasma sTGFβR3 could serve as a novel diagnostic biomarker for identifying patients with advanced tumor stages. A novel pro-tumoral mechanism of TGFβR3 downregulation via C5a-activated tumor-promoting macrophages was revealed. Therapeutic potentials involving the applications of sTGFβR3 and C5a inhibitor may represent a new approach to treating HCC patients.

## Figures and Tables

**Figure 1 cancers-13-01503-f001:**
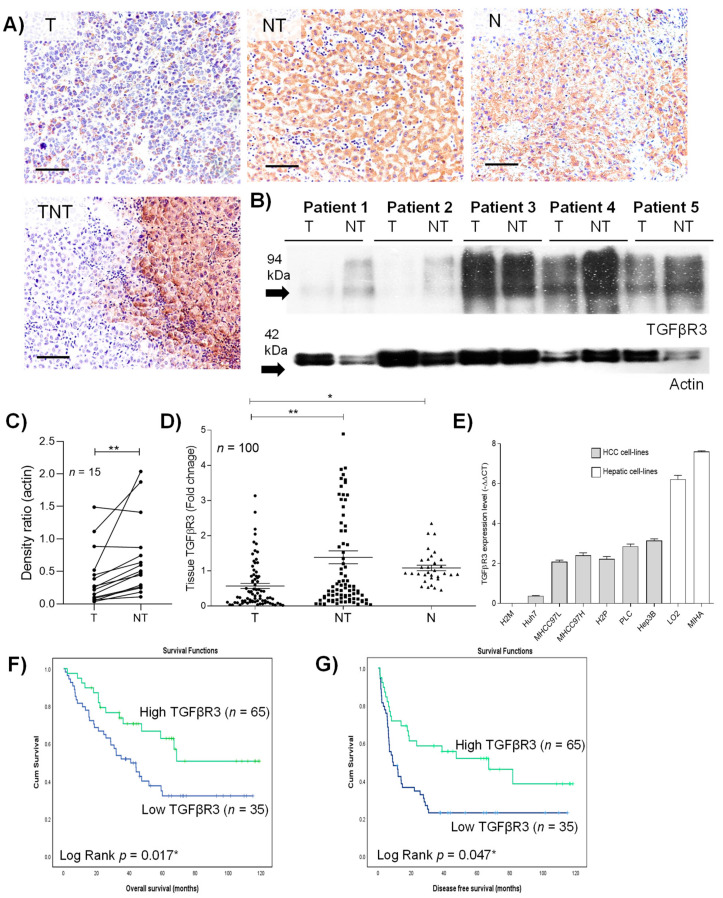
Clinical characteristics of transforming growth factor beta type III receptor (TGFβR3) in hepatocellular carcinoma (HCC) tumoral tissue. (**A**) Representative image of immunohistochemical staining of TGFβR3. (**B**) Western blotting analysis of the protein level of TGFβR3. (**C**) Densitometric quantification of Western blotting relative to actin. (**D**) Quantitative-PCR analysis of transcript level of TGFβR3. (**E**) Transcript analysis of TGFβR3 in hepatocyte and HCC cell-lines. Kaplan–Meier analysis of (**F**) overall survival and (**G**) disease-free survival in HCC patients associated with the expression level of tumoral TGFβR3 transcript. T: tumor, NT: tumor-adjacent non-tumor, N: normal liver, TNT: intra- and peri-tumor; Scale bar: 50 μm; * *p* < 0.05, ** *p* < 0.01. Error bar indicated SEM. (Unpaired *t*-test for Figure 1C,D) (Paired *t*-test for Figure 1E) (Log rank test for Figure 1F,G).

**Figure 2 cancers-13-01503-f002:**
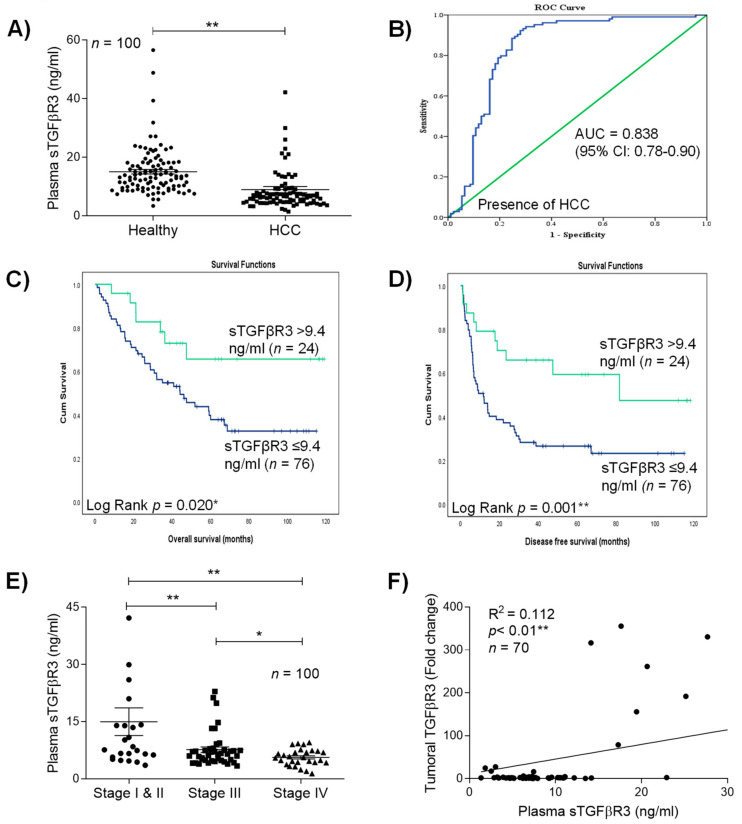
Clinical significances of sTGFβR3 down-regulation in HCC plasma. (**A**) ELISA analysis of plasma sTGFβR3 in patients. (**B**) Receiver operation characteristics (ROC) curve analysis. The clinical values were accessed by differentiation of 100 preoperative HCC patients from 100 healthy individuals. The AUC was 0.838 ± 0.032 (*p* < 0.01). (**C**,**D**) Kaplan–Meier analysis of overall survival and disease-free survival in HCC patients associated with the plasma level of sTGFβR3. (**E**) Level of sTGFβR3 in patients with different tumor stage. (**F**) Correlation between tumoral TGFβR3 and plasma sTGFβR3. * *p* < 0.05, ** *p* < 0.01. Error bar indicated SEM. (Unpaired *t*-test for Figure 2A, E) (Chi-square test for Figure 2F) (Log rank test for Figure 2C,D).

**Figure 3 cancers-13-01503-f003:**
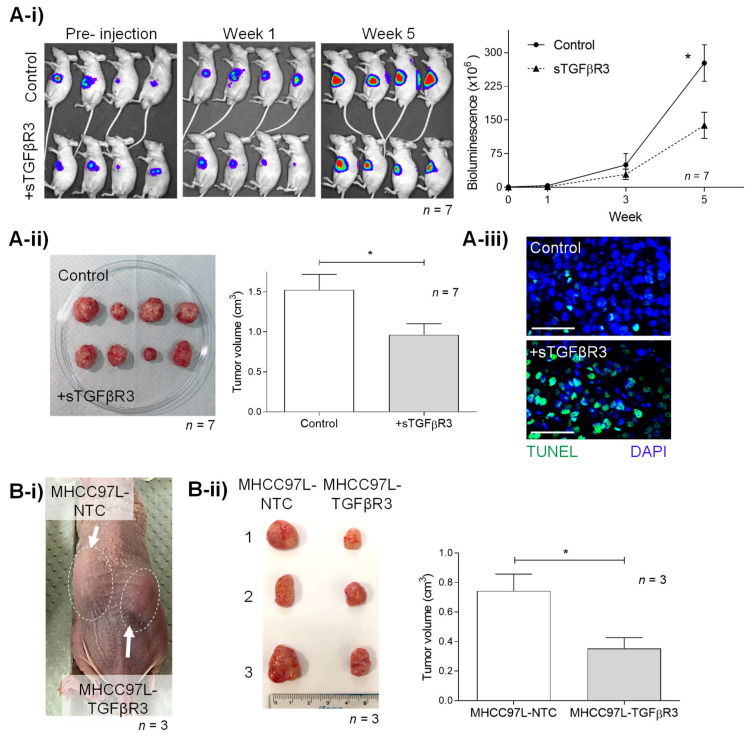
In vivo study of tumor-suppressive roles of (**A**) sTGFβR3 and (**B**) TGFβR3. (A) Male athymic mice bearing orthotopically grafted MHCC97L-Luciferase tumoral were injected with PBS (negative control) and 25 μg sTGFβR3 (*n* = 7) peritoneally weekly. (**A-i**) Monitoring of in situ tumor growth by Xenogen IVIS before and 1–5 week after injection with measurements of mean in vivo liver tumor bioluminescence of each group over time. Bioluminescent signals were quantified as photons/s at each imaging timepoint. (**A-****ii**) Following euthanasia, tumor volume was examined and measured in each mouse. (**A-iii**) TUNEL staining of liver tumor tissues in the treatment and control group. (**B-i**) Subcutaneous tumor model setup of tumor nodule induced by MHCC97L-NTC (control) on the left flank and MHCC97L-TGFβR3 (knock-in) on the right flank in each mouse (*n* = 3). (**B-ii**) Tumor volume was examined and measured after scarification in week 4. * *p* < 0.05. Scale bar: 100 μm. Error bar indicated SEM. (Unpaired *t*-test).

**Figure 4 cancers-13-01503-f004:**
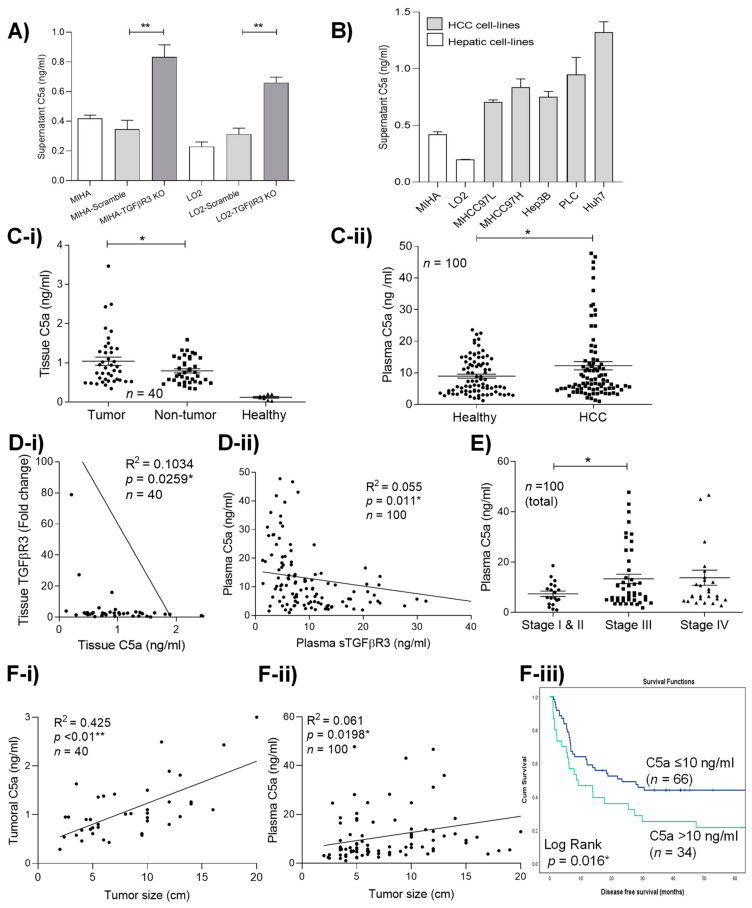
Loss of TGFβR3 induced the upregulation of C5a in HCC. (**A**) TGFβR3 transcript analysis of two non-HCC hepatocyte cell lines, MIHA and LO2, knock-out by CRISPR/Cas9 (MIHA-TGFβR3 KO) and (LO2-TGFβR3 KO). Level of secretory C5a in (**B**) hepatocyte and HCC cell lines. (**C-i**) Tumoral, adjacent non-tumoral and healthy liver tissue, (**C-ii**) healthy and HCC patients’ plasma. (**D-i**) Correlation analysis of tissue TGFβR3 and C5a. (**D-ii**) Correlation analysis of plasma C5a and sTGFβR3. (**E**) Level of plasma C5a HCC patients with different tumor stages. Correlation analysis of tumor size with the level of (**F-i**) tumoral and (**F-ii**) plasma C5a. (**F-iii**) Kaplan–Meier analysis of disease-free survival in HCC patients associated with plasma C5a. * *p* < 0.05, ** *p* < 0.01. Error bar indicated SEM. (Unpaired *t*-test for Figure 4A,B,C,E) (Chi-square test for Figure 4D,Fi-ii)(Log rank test for Figure 4Fiii).

**Figure 5 cancers-13-01503-f005:**
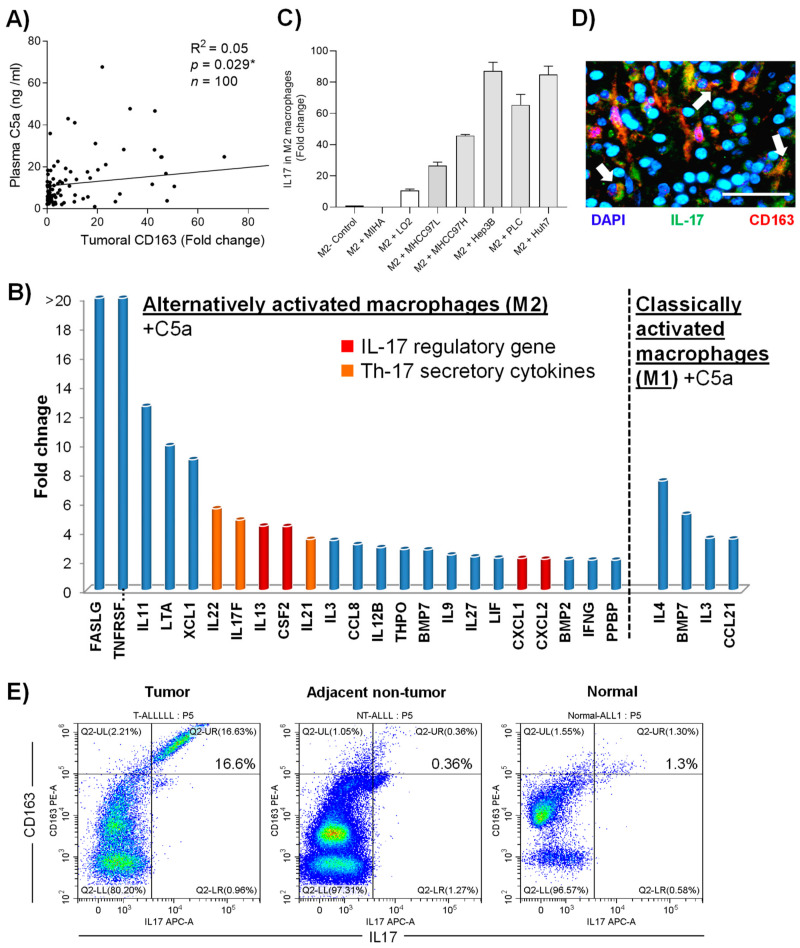
The protumor functions of C5a via M2 macrophages activation. (**A**) Correlation between plasma C5a and scavenger receptor (SA) expression level in the tumoral tissues. (**B**) cDNA expression array analysis of M1 and M2 macrophages treated with recombinant C5a compared to untreated group. (**C**) Expression of IL-17 in M2 macrophages after incubation of HCC-conditioned medium. (**D**) Double immunofluorescence image of a clinical tumoral section labeled with antibodies against IL-17 (green) and CD163 (red). (**E**) Expression of surface CD163 and intra-cellular IL-17 in total CD45+ population of tumoral, adjacent non-tumoral tissue and healthy liver tissue isolated from HCC patients measured by flow cytometry. Scale bar: 100 μm. Error bar indicated SEM. (Chi-square test for Figure 5A)(Unpaired *t*-test for Figure 5C).

**Table 1 cancers-13-01503-t001:** Correlation between TGFβR3 and clinicopathologic parameters.

	TGFβR3 Expression in Tumor with Respect to Adjacent Non-Tumor Tissue (*n* = 100)		
Clinical Parameter	Fold Change ≤ 2	Fold Change > 2	*p* Value
Gender (n)			0.794
Male	50	29	
Female	14	7	
Age (years) #			0.1
≤60	35	16	
>60	25	24	
Hepatitis B carrier			1.0
Positive	43	25	
Negative	20	12	
AFP(ng/mL)			0.014 *^
Abnormal (>20)	45	17	
Normal (<20)	17	21	
Aspartate transaminase (U/L)			0.33
Abnormal (>40)	41	21	
Normal (≤40)	21	17	
Venous infiltration			0.086
Present	40	16	
Absent	24	20	
Size of tumor (cm)			0.239
Large (>5)	38	18	
Small (≤5)	24	20	
No. of nodules (n)			0.806
Multiple (>1)	22	15	
Single (1)	40	23	
UICC grade			0.010 **^
Early stage (1,2)	19	16	
Late stage (3,4)	19	46	
Edmonson grade (differentiated)			0.003 **^
Well	9	6	
Moderately	32	25	
Poorly	23	5	
AJCC grade (stage)			0.048 *^
I	15	19	
II	23	12	
IIIA	20	11	
Ex- and intra hepatic recurrence			0.48
Present	37	19	
Absent	25	19	

Analyzed by Fisher’s exact test. * *p* < 0.05, ** *p* < 0.01. ^ Negative correlation, i.e., Pearson’s R value < 0. # Median age of TGFβR3 expression was 57.5 for fold change ≤2 group and 56 for fold change >2.

**Table 2 cancers-13-01503-t002:** Correlation between plasma sTGFβR3 and clinicopathologic parameters.

	Plasma sTGFβR3 in HCC Patients (*n* = 100)		
Clinical Parameter	≤9.4 ng/ml	>9.4 ng/ml	*p* Value
Gender (n)			0.554
Male	61	20	
Female	15	3	
Age (years)			0.631
≤60	44	15	
>60	32	8	
Hepatitis B carrier			0.447
Positive	66	21	
Negative	10	1	
AFP (ng/mL)			0.141
Abnormal (>20)	46	9	
Normal (<20)	29	13	
Bilirubin level (μmol/l)			<0.001 **
Abnormal (>20)	11	14	
Normal (≤20)	65	9	
Aspartate transaminase (U/L)			0.11
Abnormal (>40)	52	20	
Normal (≤40)	24	3	
Alanine transaminase (U/L)			0.81
Abnormal (>45)	46	13	
Normal (≤45)	30	10	
Size of tumor (cm)			0.012 *^
Large (>5)	55	10	
Small (≤5)	20	13	
No of nodules (n)			0.335
Multiple (>1)	32	7	
Single (1)	42	16	
UICC grade			0.004 **^
Early stage (1,2)	12	11	
Late stage (3,4)	63	12	
AJCC grade (stage)			0.017 *^
I	11	8	
II	32	6	
IIIA	24	3	
Ex- and in-tra hepatic recurrence			0.481
Present	46	12	
Absent	30	11	

Analyzed by Fisher’s exact test. * *p* <0.05, ** *p* < 0.01. ^ Negatively correlation, i.e., Pearson’s R value < 0. # Median age of **s**TGFβR3 expression was 57 for ≤9.4 ng/mL group and 56.5 for >9.4 ng/mL.

## Data Availability

The data presented in this study are available on request from the corresponding author.

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
