# Peer review of "Type III TGF-β Receptor Down-Regulation Promoted Tumor Progression via Complement Component C5a Induction in Hepatocellular Carcinoma"

_cancers, 2021, doi:10.3390/cancers13071503_

Round 1

Reviewer 1 Report

The authors have satisfactorily responded to the comments that the subject matter of this work is acceptable for publication.

Reviewer 2 Report

The authors responded to this reviewer's queries. This reviewer has no more issues to be addressed.

This manuscript is a resubmission of an earlier submission. The following is a list of the peer review reports and author responses from that submission.

Round 1

Reviewer 1 Report

The study by Yeung et al. investigated the clinical impact and tumor suppressive mechanisms of TGFβR3 in HCC. The authors found that TGFβR3 suppressed tumor progression and decreased expression resulted poor prognosis in HCC patients through up-regulation of tumor promoting complement C5a.

Following are some concerns which need to be addressed by the authors:

  1. Please clarify the abstract. The objectives should be clearly addressed.
  2. Include a better rationalization for choice of complement component C5a.
  3. In Figure 4A, entire experiment is performed with two non-HCC hepatocyte cell lines MIHA and LO2. To generalize the finding and to eliminate the possibility that this is not the cell phenomena, authors need to use more HCC cell lines in these experiments.
  4. Since loss of TGFβR3 can induce the up-regulation of C5a in vitro, it is highly recommend to adding some data about the effect of TGFβR3 on C5a expression in vivo.

Reviewer 2 Report

I read with interest the manuscript entitled “Type III TGF-β receptor down-regulation promoted tumor progression via complement component C5a 4 induction in hepatocellular carcinoma” by Wai Ho Oscar Yeung et al. The authors explore the role of TGFβR3 down-regulation in hepatocellular carcinoma, finding correlation with poor prognosis and tumor stage of the disease; the authors establish a concentration <9.4 ng/mL as a potential diagnostic and prognostic biomarker, and in orthotopic mouse models they observe a 37% reduction of tumor burden after continuous treatment with sTGFβR3, finding the up-regulation and macrophage activation mediated by C5a, a complement component, in TGFβR3 knockout mice.

The manuscript is interesting and the data are well presented, conclusion are supported by well designed experiments.

This reviewer has only a few comments to be addressed by the authors.

In Table S1 authors describe the analyzed case series. The age range is very large, comprising a 3 years old too; are the authors sure to include these patients, knowing that children liver pathology is different from the adult one? Moreover, HCC is very rare in children <12 years old, are the authors sure to not include hepatoblastomas?

The case series patients is composed by 29% stage IV patients, and all were addressed to surgical treatment. Nonetheless, only 1% of the patients were evaluated as grade C through Child-Pugh. It seems so rare in a “randomly collected” case series.

Paragraph 4.1: how the tissue by healthy donors was provided? Did this people receive a biopsy for any liver diagnostic purpose? Please provide an approval code for the present study.

Line 337: was the concentration of C5a 2.5ug/mL? How the authors selected this concentration?

Minor concerns

Line 11: please rephrase the sentence. Hepatocellular carcinoma is not the unique cancer of the liver.

Line 14-15: the sentence is not clear; please specify that patients with poor prognosis were a subset of patients diagnosed with HCC, if it is so.

Figure 1E-F: please remove decimals from the table at X axis - months

Reviewer 3 Report

The article entitled  “Type III TGF-β receptor down-regulation promoted 3 tumor progression via complement component C5a 4 induction in hepatocellular carcinoma” by  Wai Ho Oscar Yeung is an interesting article and of scientific significance.

In this article, the authors described the clinical significance of  TGF-β receptor type III (TGFβR3) in tumor suppression in HCC and described its role in detail. Authors have Successfully used different model systems human (patient) as well in mice and other in-vitro models and systematically presented their observations. However, authors should address the following to further strengthen their study.

  1. Make “Simple summary” shorter and to the point
  2. Organize the Abstract section as Background, Aim/ hypothesis, Martials /method, results, and conclusion.Currently, the body of the abstract is jumbled and connecting sentences are missing.,  “This study aimed to investigate the clinical 25 impact and tumor-suppressive mechanisms of TGFβR3 in HCC. Clinical specimens were obtained 26 from 100 HCC patients who underwent curative resection. TGFβR3 and its soluble form (sTGFβR3) 27 were quantified by qPCR and ELISA, respectively, and their clinical correlations were analyzed.” Authors should explain why and how you obtained and used human and mice data),
  3. Graphical abstract 9lower panes) needs to be clearer and should be presented in an easy to follow sequence.  
  4. In the” introduction” section, explain what is new in this current and why this study is clinically important.
  5. Explain more about the clinical study and patient samples, like what is the median age of the patients in a particular group. What is the registration/license number for the study?
  6. In Fig 1A, provide images of similar magnification. Also,  densitometric analysis of the western blot is important to conclude.
  7. Please provide the  “N”  for each observation, in some cases, it is not clear, like in   Fig3Bii.
